The structure and diversity of freshwater diatom assemblages from Franz Josef Land Archipelago: a northern outpost for freshwater diatoms

Pla-Rabés Sergi 1 sergiplarabes@gmail.com
Hamilton Paul B. 2
Ballesteros Enric 3
Gavrilo Maria 4
Friedlander Alan M. 5 6
Sala Enric 6
1 CREAF , Cerdanyola del Valles, Barcelona , Spain
2 Canadian Museum of Nature, Research and Collections , Ottawa, Ontario , Canada
3 CEAB, CSIC , Blanes, Girona , Spain
4 National Park Russian Arctic , Archangelsk , Russia
5 Fisheries Ecology Research Laboratory, Department of Biology, University of Hawaii , Honolulu, Hawaii , USA
6 National Geographic Society , Washington, D.C. , USA
Gupta Vijai Kumar
Electronic publication date: 2016 Feb 18
Publication date: 2016
Volume: 4
Electronic Location ID: e1705
Received 2015 Oct 26; Accepted 2016 Jan 28
Copyright: ©2016 Pla-Rabés et al.
Copyright year: 2016
Copyright holder: Pla-Rabés et al.
License: This is an open access article distributed under the terms of the Creative Commons Attribution License, which permits unrestricted use, distribution, reproduction and adaptation in any medium and for any purpose provided that it is properly attributed. For attribution, the original author(s), title, publication source (PeerJ) and either DOI or URL of the article must be cited.
License URL: https://creativecommons.org/licenses/by/4.0/

Keywords: Arctic, Climate change, β-diversity, Diversity, Indicator value, Indval, Ponds, Streams

Funding: National Geographic, Blancpain, Davidoff Cool Water, the Russian Arctic National Park Russian Geographical Society State Nature Reserve—Franz Josef Land Canadian RAC Funding for this work was provided by National Geographic, Blancpain, Davidoff Cool Water, the Russian Arctic National Park, the Russian Geographical Society, and the State Nature Reserve—Franz Josef Land. A supporting Canadian RAC (2015–2017) grant to PBH was used to complete taxonomic SEM and LM investigations. The funders had no role in study design, data collection and analysis, decision to publish, or preparation of the manuscript.

==============================
We examined diatom assemblages from 18 stream and pond samples in the Franz Josef Land Archipelago (FJL), the most northern land of Eurasia. More than 216 taxa were observed, revealing a rich circumpolar diatom flora, including many undescribed taxa. Widely distributed taxa were the most abundant by cell densities, while circumpolar taxa were the most species rich. Stream and pond habitats hosted different assemblages, and varied along a pH gradient. Diatoma tenuis was the most abundant and ubiquitous taxon. However, several circumpolar taxa such as Chamaepinnularia gandrupii, Cymbella botellus, Psammothidium sp. and Humidophila laevissima were also found in relatively high abundances. Aerophilic taxa were an important component of FJL diatom assemblages (Humidophila spp., Caloneis spp. and Pinnularia spp.), reflecting the large and extreme seasonal changes in Arctic conditions. We predict a decrease in the abundance of circumpolar taxa, an increase in local (α-) freshwater diatom diversity, but a decrease in regional diversity (circumpolar homogenization) as a result of current warming trends and to a lesser extent the increasing human footprint in the region.

Introduction

The Arctic is warming faster than any other region on the globe, which imposes new challenges to the resilience of this region and its diversity. Freshwater ecosystems are particularly sensitive to global and local stressors such as climate change, atmospheric pollution, alien species introductions, and resource development, which can all alter biogeochemical cycles. Much of the Arctic is still poorly explored, hence little is known about the impact of climate change on northern freshwater and marine ecosystems (e.g., Chernova et al., 2014). There is on-going international interest in expanding monitoring programs on climate change, particularly in the polar regions (UNESCO, 2009; AMAP, 2015). This will become increasingly more important as resource development and tourism in the Arctic is expected to increase dramatically in the coming decades.

Arctic ponds and lakes can be considered as sentinels of the changing Arctic (Williamson et al., 2008). Their freshwater living biota, mainly diatoms (Bacillariophyceae) and cyanobacteria, are the most abundant and diverse photosynthetic groups found in these habitats. Paleolimnological studies using freshwater diatoms across the Arctic have documented past climatic changes (e.g., Smol et al., 2005). The most striking change, which is still ongoing, has been the expansive increase in diatom productivity and diversity since the 1950s (Gajewski, Hamilton & McNeely, 1997). Despite these rapid changes, we have a limited understanding of the stability (i.e., fluctuations in the occurrence and abundance of species through time) of these freshwaters systems. In addition to changing climate, the expanding human footprint in the Arctic has resulted in increased contamination by heavy metals such as mercury and arsenic (Chetelat et al., 2014). Questions about biological stability, including native species loss and introduction of invasive species have yet to be addressed.

The Franz Josef Land (FJL) Archipelago is the most northerly land in the Arctic region of Eurasia. All islands in the archipelago belong to the specially protected area of Franz Josef Land state federal wildlife refuge (corresponding IUCN category IV) and are managed by the Russian Arctic National Park. High Arctic exploration, mining, fishing, and tourism, particularly to the remote northern islands, has always been limited due to logistical constraints. Consequently, little is known about the ecology and distribution of the FJL freshwater diatom flora. Nevertheless, at the end of the 19th century, the Austro-Hungarian Tegetthoff expedition (1872–1874) sampled marine bottom sediments and Grunow (1884) noted several diatom species in his monograph, but only a few freshwater diatoms were reported. Later in that century the Jackson–Harmsworth Expedition to FJL (1894–97) collected about 20 samples, mainly from Cape Flora, but also from Cape Neal and Bell Island, including a mud sample from an off-shore drifting ice-floe (Cleve, 1898). A total of 101 diatom morphotypes were observed, including 27 freshwater taxa identified in the ice-floe sample. During the first half of 20th century, Shirshov (1935) sampled four lakes in FJL and compared their algal flora with lakes at Novaya Zemlya, a large Russian island ∼500 km to the southeast of FJL. Diatom floras from the high Arctic (FJL, Spitsbergen, Beeren Island, Jan Mayen and East and West Greenland) differed from floras in the lower Arctic region (e.g., Siberia and Lapland) (Cleve, 1898).

Since the 1950s numerous surveys have increased our knowledge on the rich Arctic diatom flora (Foged, 1953; Foged, 1955; Foged, 1958; Foged, 1964; Foged, 1973; Foged, 1974; Foged, 1977; Foged, 1981; Lange-Bertalot & Metzeltin, 1996; Metzeltin & Witkowski, 1996; Lange-Bertalot & Genkal, 1999; Antoniades et al., 2008), but there is only one recent study on diatoms from FJL (Zauer, 1963).

New species are frequently described as locally endemic (i.e., Van De Vijver, Beyens & Lange-Bertalot, 2004), but are quite often later reported from other Arctic localities. Hence, several studies support the idea of a circumpolar diatom distribution (Lange-Bertalot & Metzeltin, 1996; Metzeltin & Witkowski, 1996; Lange-Bertalot & Genkal, 1999; Antoniades et al., 2008). Drifting icebergs and ice-floes may act as vectors in the circumpolar distribution of microorganisms (Cleve, 1898). In addition, the ability of some diatoms to survive aerial voyages by wind and bird transportation are also possible dispersal mechanisms (Kristiansen, 1996). Consequently, there is a need to explore more Arctic sites to understand the biogeography of circumpolar microorganisms.

The objective of this research is to document parts of the diatom flora of FJL, a poorly studied region in the high Arctic in order to establish a baseline for future studies. We also examine biogeographical connections with Greenland, Siberia, Spitsbergen, and northern Canada. Finally, we propose a hypothesis-driven theoretical framework on the effects of recent and past climate warming scenarios on species richness for future studies and investigate future scenarios for conservation of circumpolar biodiversity.

Methods

Study area and sites

FJL is a large archipelago that occupies an area of 16,000 km2 and consists of 192 islands. The archipelago was formed together with the Spitsbergen Archipelago by the uplift and broken margin of the Barents Shelf (Solheim, Evgeny & Heintz, 1998). FJL is basically a remnant of the basaltic plateau fractured by tectonic faults and covered by glaciers (over 85%), with the highest elevation at ca. 600 m. Most of the islands are small, only nine exceed 500 km2 with large ice-free areas of 80–500 km2 on only four islands.

FJL lies within the high arctic climatic zone, with an average annual air temperature of −13.2 °C. July and August are the only two months with average air temperatures above zero (+0.7 °C and +0.1 °C, respectively) (VNIIGMI-MCD data accessed at http://oopt.aari.ru). Periods with above freezing average daily temperatures last from 2 months in the southern portion of FJL to 1.5 months in the north. Rainfall averages 250–300 mm year−1 with only 40 mm during warm months. On average, FJL is much colder than Spitsbergen, which has an annual air temperature −7 °C, and four months with air temperatures above zero (Førland et al., 2011), warmer than the Canadian high Arctic islands (e.g., average annual air temperature in Eureka, Nunavut −19.9 °C, additional data can be found in http://www.eureka-nt.climatemps.com/temperatures.php).

The hydrographical network of FJL is not well developed; most of the streams are intermittent, while all rivers are completely frozen in winter. There are over 850 lakes and ponds in FJL but only 10 of them are deeper than 1 m (i.e., not frozen to the bottom M Gavrilo field observation). Principal water sources are melting snow and glaciers. Thus, most of freshwater habitats are very harsh, having short growing season and are only temporary.

Approval to conduct field studies was granted by the Russian Federation Ministry of Education and Science Approval Ref. No. 14-368 of 06.05.2013 permission to the Russian Arctic National Park.

Diatoms analysis

During the 2013 National Geographic Pristine Seas expedition to FJL, 18 samples from freshwater habitats (2 lakes, 7 ponds, and 9 streams) were sampled from different islands across the archipelago (Fig. 1 and Table 1). Scrapings were done on rock surfaces, which consisted of a mixture of mosses and cyanobacteria biofilm. In the field, samples were fixed and preserved in 3% formaldehyde.

Figure 1 Diatom samples localities in Franz Josef-Land Archipelago.

Orange dots denote stream samples, and green dots pond samples. Numbers are sample codes from Table 1.

Table 1 Sample localities, geographic coordinates, habitat type and sampling date.

Code	Locality	NN	EE	Habitat	Date	S	ExpH	Simpson	EF-1	EF-2	
FJ01	Cape Flora, Northbrook Island	79.9453	50.1295	Pond 1	5 August	37	5.8	3.3	0.156	0.089	
FJ02	Cape Flora, Northbrook Island	79.9453	50.1295	Pond 2	5 August	30	10.3	6.7	0.345	0.224	
FJ03	Cape Flora, Northbrook Island	79.9453	50.1295	Stream	5 August	9	1.9	1.5	0.212	0.165	
FJ04	Cape Flora, Northbrook Island	79.9453	50.1295	Pond 3	5 August	35	10.8	7.4	0.307	0.211	
FJ05	Cape Flora, Northbrook Island	79.9453	50.1295	Pond 4	5 August	34	12.4	8.4	0.365	0.248	
FJ06	Tikhaya Bay, Hooker island	80.3362	52.7758	Pond 1	31 July	64	26.8	14.1	0.418	0.220	
FJ07	Tikhaya Bay, Hooker island	80.3362	52.7758	Pond 2	31 July	40	9.4	5.6	0.234	0.139	
FJ08	Wilczeck Land	80.6109	61.9859	Stream	15 August	18	8.7	7.0	0.483	0.388	
FJ09	Mabel Island	80.0193	49.3579	Nearshore stream	25 August	31	5.7	3.3	0.183	0.105	
FJ10	Mabel Island	80.0193	49.3579	Uphill stream	25 August	21	7.1	4.8	0.340	0.227	
FJ11	Nansen Island	80.5502	54.1329	Glacier stream	23 August	36	13.3	8.7	0.369	0.243	
FJ12	Mabel Island	80.0111	49.3791	Small lake 1	21 August	36	12.1	6.8	0.337	0.189	
FJ13	Mabel Island	80.0111	49.3791	Small lake 2	21 August	36	13.5	8.4	0.376	0.234	
FJ14	Wilton Island	80.5613	54.3557	Stream	23 August	64	25.2	15.0	0.393	0.234	
FJ15	Cape Tegethoff, Hall Island	80.0926	58.0584	Nearshore pond	11 August	59	30.8	17.3	0.522	0.294	
FJ16	Kuhn Island	81.1047	58.3721	Small stream	9 August	48	18.2	9.9	0.379	0.206	
FJ17	Alger Island	80.3529	56.1413	Small stream 1	14 August	36	9.4	5.2	0.261	0.144	
FJ18	Alger Island	80.3820	55.7698	Small stream 2	14 August	35	11.7	7.8	0.334	0.222	
Notes.

NN latitude N

EE longitude E

S species richness

ExpH exponential of Shannon Entropy

Simpson Simpson diversity index

EF-1 evenness of order 1

EF-2 evenness of order 2 (Jost, 2010)

Diatom samples were processed by hot digestion using hydrogen peroxide (33% H2O2) and HCL (2 ml 1 M). Through a series of dilutions, the peroxide and acid were removed. Subsequently, the samples were dried onto cover glass and mounted in Naphrax® (R.I. = 1.7, Brunel Microscopes Ltd., UK) on a microscope slide following the method described in Battarbee et al. (2001). Five hundred diatom valves per slide sample were identified and counted across random transects at 1,000× magnification using a Zeiss Axio Imager A1 microscope (Carl Zeiss Inc., Germany) equipped with a 100× objective (Zeiss Plan-Apo 1.4 numeric aperture) and differential Interference contrast optics. Scanning electron micrographs were taken using a FEI Quanta 2000-F-SEM (10 kV, WD 10 mm).

Diatom identifications were based on contributions by Lange-Bertalot & Metzeltin (1996), Metzeltin & Witkowski (1996), Lange-Bertalot & Genkal (1999), Krammer (2000), Lange-Bertalot (2001b); Lange-Bertalot (2001a), Krammer (2002), Krammer (2003), Van De Vijver, Beyens & Lange-Bertalot (2004), Antoniades et al. (2008), Kulikovskiy et al. (2010), Zimmermann, Poulin & Pienitz (2010), Lange-Bertalot, Bak & Witkowski (2011) and Levkov (2013).

Data analysis

Detrended correspondence analysis (DCA) was conducted to estimate the length of the gradient on the square-root-transformed diatoms abundance data. Due to the large length of the gradient (>5 SD), a Principal Coordinates Analysis (PCO) was used based on Hellinger distances to reveal the structure among the sampled diatom assemblages (Legendre & Legendre, 1998). Only diatoms with abundances >1% in at least one sample were included in the PCO, which was performed using CANOCO 5.0 (Ter Braak & Šmilauer, 2012).

Diatom assemblage diversity was calculated in terms of species richness (S) and the exponential of Shannon’s entropy [exp(H)], which is more appropriate than classical diversity indices for assessing statistically independent measures of alpha (α) and beta (β) diversity (Jost, 2007). For each habitat type (lentic and lotic), diversity (D) was partitioned into α, β, and gamma (γ; total diversity) components following Jost (2007). Beta diversity was used as a measure of the differences in the species composition between habitat types. Evenness was calculated using two diversity orders. The first, of order 1, is based on Shannon entropy (EF1) and the second, of order 2 (EF2) is based on Simpson’s diversity metric (Jost, 2010). Hence, EF2 is an approach to the proportion of the dominant species in the assemblage, whereas EF1 also takes into account all the common species in the assemblage.

In order to recognize a discontinuous subset of samples, a K-means clustering with the optimal number of groups chosen according to the Calinski-Harabasz pseudo-F-statistic was performed (Legendre & Legendre, 1998). The ecological indicator values for each diatom taxon were identified to assess the ecological preferences of individual species for each of the clusters. The Indval-index was employed to control the effects of unequal size of groups (De Cáceres & Legendre, 2009).

All analyses of diversity metrics were performed using the R statistical program (R v. 3.0.2.; http://www.r-project.org/) and additional functions from the R-packages “vegetarian” v. 1.2 , “vegan” v. 2.2-1 and “indicspecies” v. 1.7.4 to calculate Indval-index.

Results

A total of 216 benthic diatom taxa belonging to 44 genera were identified within 18 samples from FJL. Seventy-two taxa occurred only in lentic (ponds/lakes) environments, 42 occurred only in lotic (streams) environments, while the remaining taxa were found in both habitats. The checklist of all taxa with their means, highest abundances, and biogeographic distributions (circumpolar versus widely distributed) are found in Appendix S1. Widely distributed species, or those with broader distribution than circumpolar, dominated the samples by cell numbers, but more circumpolar taxa dominated species richness (Appendix S1). In lotic ecosystems, the following widely distributed taxa were dominant: Diatoma problematica, Fragilaria capucina (complex), Rossithidium petersenii, Diatoma tenuis, Eucocconeis laevis, Psammothidium marginulatum, Nitzschia homburgensis, Meridion circulare, Achnanthidum minutissimum sensu lato, Psammothidium kryophilum and Hannaea arcus. Taxa such as P. marginulatum and A. minutissimum sl. are composed of a mix of widely distributed and circumpolar taxa. Four other circumpolar taxa had abundances >9%: Encyonema fogedii (complex), Cymbella botellus, Psammothidium sp1, and Cymbella cleve-eulerae. Several widely distributed taxa showed high abundances in lentic environments, including: Nitzschia alpina, Tabellaria flocculosa, Encyonema silesiacum, Naviculadicta digitulus, and Humidophila gallica. In contrast, circumpolar taxa such as Chamaepinnularia gandrupii, Fragilaria cf. perminuta, Nitzschia cf. sublinearis, and Hippodonta arctica had lower abundances in lentic habitats (Fig. 2). Two widely distributed taxa showed similar patterns of dominance in both lotic and lentic habitats: Diatoma tenuis and Hygropetra balfouriana. A number of taxonomically unidentified forms were also observed (Appendix S1).

Figure 2 Dot plot of FJL diatom assemblages for all samples sites and diatoms with higher abundance (>2%).

Species are sorted across PCO axis 2 scores (pH gradient). Samples are sorted across the PCO axis 1 (from ponds to streams). Sites in blue denotes stream habitat. Symbol size indicates the relative abundance of each taxa for each site. Symbol colour indicates Indval value for species in each K-means cluster. Symbol shape denotes species geographic distribution.

Ponds supported 82 circumpolar taxa and streams 62. Mean taxa richness per sample was 41 (±12 sd) for ponds and 33 (±16 sd) for streams, although this difference was not significant (t-test = 1.20, p > 0.05). A similar pattern was observed for diversity metrics (Fig. 3). β-diversity showed large differences between stream and pond habitats (β-diversity of order 1; Hβ1=3.4 for streams and Hβ1=2.8 for ponds). Stream samples showed higher variability compared to ponds (Fig. 4).

Figure 3 Stream and pond (including lake) sample diversity metrics boxplots.

Evenness-F1 and Evenness-F2 are evenness factor of order 1 and of order 2 following Jost (2010).

Figure 4 PCO results.

Ellipses are grouping results of K-means cluster analysis. Only the 25 species with highest correlation with one of the two main axes are shown. Numbers in red denotes K-means cluster. Squares denote stream samples and circles pond samples. Full species names are reported in Appendix S1.

One stream sample from Cape Flora was dominated by Diatoma problematica (80% of the assemblage), and was removed for further analysis (FJ03, Table 1 and Fig. 2). K-means cluster and PCO analysis showed five main assemblages (excluding FJ03 sample). The first PCO component explained 18.9% of the variability, and showed some overlap between ponds/lake and stream diatom assemblages. On the negative end of this first component, the species with higher correlation were R. petersenii, P. marginulatum and Cymbopleura stauroneiformis (Fig. 4); Caloneis holarctica, Nitzschia cf. sublinearis, and Stauroneis subgracilis were at the positive end of Component 1 (Fig. 4). Additionally, stream diatom assemblages were separated in two groups along the second PCO axis, which explained an additional 13.6% of diatom assemblage variability (Fig. 4). Negative scores were mainly associated with northern sites (Nansen, Wilton, and Kuhn islands), with Eunotia bilunaris, Neidium bisulcatum, Humidophila ingeiiformis, and H. laevissima showing the highest correlation. On the positive end of PCO 2, the species with the highest correlation were Amphora copulata, E. fogedii (complex), Psammothidium bioretii, Encyonema cf. spitsbergense, and E. laevis (Figs. 2 and 4). The third PCO axis explained an additional 10.1% of variability, which separated lake samples (cluster 4) from the other habitats. Widely distributed species such as Encyonema reichardtii, E. silesiacum, Staurosirella pinnata, Nitzschia alpina, and N. digituloides had the greatest influence in the lake cluster.

In order to further identify representative species for each cluster, species indicator values (Indval) were applied. Cluster 4 (lakes) was characterised by more widely distributed species, which were highly correlated with PCO axis 3. Cluster 2, which had the highest species richness and diversity, was characterised by aerophilic Arctic species: Caloneis fusus, Humidophila paracontenta, Caloneis fasciata, H. laevissima, and Luticola paleoarctica, as well as a high abundance of C. gandrupii. In addition, Nitzschia homburgensis, which has a broad biogeographic distribution (Fig. 2), was also found in cluster 2. Cluster 3 was characterized by circumpolar species such as C. gandrupii, C. holartica, N. cf. sublinearis, Nitzschia sp.A, and Sellaphora rectangularis sensu lato (Fig. 2). Cluster 5 consisted mainly of C. stauroneiformis, Gomphonema nathortsii, Psammothidium sp1, and Eunotia scandiorussica. Cluster 1 consisted of three cosmopolitan species, A. copulata, Encyonema laevis, and Nitzschia cf. acidoclinata, along with one widely distributed species—E. fogedii (complex). In addition, some circumpolar species such as Gomphonema lapponicum, Amphora dusenii, and Amphora spitzbergensis only occurred in samples from this cluster (Fig. 2).

Although, diversity measures did not show significant differences between stream and ponds, statistical differences were observed among clusters. Cluster 2 (Fig. 5) showed the highest diversity and richness per sample. This group consisted of two stream samples (Kuhn and Wilton islands) and three pond samples (Cape Tegetthoff, Cape Flora, and Tikhaya Bay). Measures of equitability showed that higher richness was mainly due to the codominance of “satellite” species rather than the codominance of only dominant ones. This higher richness was significant for EF1 but not for EF2 (Fig. 5).

Figure 5 Diversity metrics boxplots for each k-means clusters (see plot legend).

Evenness-F1 and Evenness-F2 are evenness factor of order 1 and of order 2 following Jost (2010).

Discussion

Diversity and distribution patterns

The reported number of diatom taxa per sample in our study was similar to previous Arctic surveys (Douglas & Smol, 1995; Laing, Pienitz & Smol, 1999; Weckstrom & Korhola, 2001; Antoniades & Douglas, 2002; Michelutti et al., 2003; Jones & Birks, 2004; Antoniades, Douglas & Smol, 2005). However, these surveys also sampled surface sediments, which could have increased the number of observed taxa per sample. Surface sediment samples often integrate a disparity of habitats over time. Our study also found many diatom morphotypes that could not be identified, illustrating that taxa still need to be described, and supporting the recognition of high diversity and richness of freshwater diatoms across the Arctic (Foged, 1981; Antoniades et al., 2008; E Pinseel et al., unpublished data). Many species (new Arctic taxa) are presently included within species complexes (e.g., A. minutissimum, N. cf. perminuta, E. cf. fogedii, P. marginulatum) or yet to be described (e.g., Psammothidium sp1).

The large compositional turnover among samples (5.3 sd in the DCA) indicates large differences in diatom assemblages among sites. The main source of diatom assemblage variability was related to habitat (PCO axis 1; lotic versus lentic). Nevertheless, two clusters (2 and 5; Fig. 4) had samples from both lotic and lentic habitats, similar to findings from Spitzbergen (E Pinseel et al., unpublished data). Small pond assemblages could be largely influenced by the arrival of taxa from streams (Antoniades & Douglas, 2002), and conversely streams with low velocity (discharge) could support lentic assemblages. Unfortunately, we do not have stream velocity or water residence time data to confirm these observations. However, characteristic taxa of stream habitats in FJL, such as R. petersenii and F. capucina (complex), were also common in FJL pond samples. This observation indicates that pond diatom assemblages in FJL may be influenced by streams, as observed in other Arctic regions (Antoniades & Douglas, 2002; E Pinseel et al., unpublished data). Therefore, the dominance of stream taxa in pond sample FJ07 suggests a large influence of stream species. The same reasoning can be applied to the pond samples in cluster 2 (FJ04, FJ06, and FJ15), but results are less evident. This cluster, showed higher richness and diversity, with a dominance of circumneutral/slightly acidic taxa (pH optima ca. 6.8–7.2) such as C. gandrupii, H. gallica, H. laevissima, P. marginulatum, N. homburgensis, and H. balfouriana (Antoniades, Douglas & Smol, 2004; Antoniades et al., 2008). On the PCO biplot, cluster 1 (Fig. 4, with only stream samples), was characterised by the presence of alkaliphilic taxa (optimum pH 8.0–8.3 in Antoniades et al. (2008)) such as, E. laevis, P. bioretii, E. fogedii (complex), A. copulata, and A. inariensis. Therefore, this second axis of diatom variability is likely correlated with a pH gradient, which is a main source of variability for diatom assemblages (Battarbee et al., 2001), with Algers, Mabel, and Wilkzec Islands potentially marking alkaline conditions with higher pH. The pH gradient should be, in part, related to the complex archipelago geology with intrusions of basalt between sedimentary bedrocks (Dibner & Fursenko, 1998). However, recent deglaciated areas tend to have increased alkalinity and pH in lakes and streams from bedrock and till exposure (Engstrom et al., 2000; Milner et al., 2007). Thus, lake age (Milner et al., 2007) could be an alternative hypothesis to explain the lower pH optima of the characteristic taxa from cluster 2, rather than changes in the site geological settings.

Interestingly, the samples from cluster 2 also showed higher diversity metrics, due to the higher equitability among the non-dominant taxa (EF1). Less stable environments exposed to physical stressors (e.g., current flow) could impact dominance by minimizing the effects of competitive exclusion and promoting species coexistence (Hughes et al., 2007) maintaining species diversity (Chesson, 2000). In Antarctic streams, environmental instability (changes in stream velocity and successive periods of freezing and thawing) promote small cell size, maintaining higher diatom richness (Pla-Rabes et al., 2013), with diatom assemblages showing higher resilience (recovery to pre-disturbance conditions; e.g., flood event) (Stanish, Nemergut & McKnight, 2011). There are indicators to suggest that the biology at some FJL sites support high environmental variability. First, cluster 2 samples are characterised by the presence of aerophilic taxa (H. laevissima, H. paracontenta, and C. fusus), which indicate potential periods of lower water availability, and possibly even desiccation. Second, species from cluster 2 showed lower pH optima (6.8–7.2 in Antoniades et al., 2008) compared to cluster 1 (7.9–8.2 in Antoniades et al., 2008), which means lower buffering capacity in cluster 2 sites, and consequently higher environmental variability. Furthermore, lower alkalinity sites would tend to be older (Milner et al., 2007), and consequently have a higher chance of accumulating species (Adler et al., 2005). Interestingly, cluster 2 sites showed large expanses of ice-free areas (Khun and Wilton Islands), and more human influence (Tikhaya Bay, Cape Tegetthoff, Cape Flora). Nevertheless, our data set did not include samples from the largest ice-free areas located on the largest islands in the archipelago (Alexandre Land and Prince George Land), which could potentially be diversity hotspots due to large biogeographic sizes.

Biogeographical patterns

The general diatom flora observed in FJL is similar to other circumpolar localities (Cleve, 1898; Lange-Bertalot & Genkal, 1999; Antoniades & Douglas, 2002; Stewart, Lamoureux & Forbes, 2005; Antoniades et al., 2008). A comparison of photomicrograph documented taxa in this study (Appendix S2) with Antoniades et al. (2008) for the Canadian Archipelago shows that 67% of the taxa observed in FJL were present in both studies. This indicates a circumpolar distribution, along with some local species selectivity due to microhabitats and regions. In an earlier study, Foged (1981) demonstrated that many more species are present in Arctic microhabitats, but often not recorded due to their rare distributions. Interestingly, a small araphid species flock representing Staurosirella, Staurosira, and Pseudostaurosira, was scarce in FJL, with only S. pinnata abundant and characteristic of pond samples in cluster 2. This was also observed in Svalbard (E Pinseel et al., unpublished data). The Staurosirella, Staurosira, and Pseudostaurosira flocks typically dominate hard substrate habitats in low productivity ponds and lakes across the Canadian and western Russian Arctic (e.g., Metzeltin & Witkowski, 1996; Laing, Pienitz & Smol, 1999; Michelutti et al., 2003; Bouchard, Gajewski & Hamilton, 2004; Paull et al., 2008). In our study of the twenty most abundant taxa, Nitzschia and Psammothidium were the most common genera with no clear associations within species flocks. These genera, along with Achnanthidium, are indicative of moss and sediment microhabitats. Moss wetlands have a diverse flora with select genera like Luticola, Humidophila, and Eunotia having unique compositions (Douglas & Smol, 1995; Van De Vijver, Van Kerckvoorde & Beyens, 2003). Luticola taxa observed from FJL were similar to the Novaya Zemlya Archipelago (Lange-Bertalot & Genkal, 1999), while different from the Canadian Archipelago (Antoniades et al., 2008). Some common Arctic stream species, such as Hannaea arcus, occurred in low abundance at FJL, which could be related to the absence of specific habitat requirements such as high current velocity (Antoniades & Douglas, 2002). Likewise, taxa from the genera Humidophila and Placoneis appeared to be species rich in FJL, which has also been observed in Svalbard (E Pinseel et al., unpublished data). In locations with more productive freshwater habitats, taxa from the family Cymbellaceae (Cymbella, Cymbopleura, and Encyonema) are noted to be abundant, with greater species richness, and widely distributed across microhabitats (Douglas & Smol, 1995; Michelutti et al., 2003). Except for C. gandrupii, currently recognized circumpolar taxa do not dominate FJL assemblages, although species richness is clearly linked to circumpolar and regionally endemic taxa (Fig. 2).

Geographic dispersal, especially with respect to a potential circumpolar flora, requires mechanisms that are inherently controlled by climate. Freshwater organisms such as diatoms could be dispersed by air as well as organisms such as birds, fishes, or mammals (Kristiansen, 1996). Interestingly, Nansen (1897) documented a similarity in diatoms and microbes in melt water pools from the Bering Strait and Greenland, and along with other observations, proposed that drifting ice could pass across the North Pole. Cleve (1898) was the first to suggest a potential circumpolar distribution of diatom species. Arctic drift ice has preferential pathways throughout the Arctic region and has been found to transport viable freshwater algae such as diatoms (Cleve, 1898; Abelmann, 1992; Pfirman et al., 1997). Therefore, drifting ice could be an important mechanism for dispersal, hence the recognition of a circumpolar flora. Additionally, airborne diatoms could also reach different Arctic regions. For example, dust storms routinely transport particulates from northern Asia to western North America (Chin et al., 2007). In this regards, population genetic studies of diatoms across the Arctic can assist in answering questions of biogeographic distribution (Hamilton, Lefebvre & Bull, 2015). Specifically, population genetics may help identify potential vectors of migration like drifting ice trajectories, atmospheric wind patterns, and bird migration pathways.

Consequently, the concept of circumpolar diatom species is applicable to a broader region including northern boreal regions from the Palearctic and Nearctic areas. For instance, the presence of Boreozonacola hustedtii recently described from a Mongolian peat-bog (Kulikovskiy et al., 2010), has been found in Siberia (Lange-Bertalot & Genkal, 1999), Lapland (Lange-Bertalot & Metzeltin, 1996), Canada (Zimmermann, Poulin & Pienitz, 2010), the Rocky Mountains USA (Bahls, 2010), and in FJL.

Future scenarios of diversity

Franz Josef Land Archipelago is a “zakaznik” Russian protected area (nature refuge corresponding to IUCN IV). As in Antarctica, climate change and an increasing human footprint are the two main threats for conservation in FJL (Chown et al., 2012). However, due to the poor knowledge of diatom distribution and ecology, it is difficult to predict future changes in productivity and diversity. Current predictions are theoretical using knowledge from other regions and ecological theory.

Arctic biodiversity has been exposed to strong selection mechanisms in harsh and fluctuating environments (glaciation/deglaciation periods) for millennia. However, at present we consider rapid climate change as the main factor impacting biodiversity in polar ecosystems (Millennium Ecosystem Assessment, 2005). In addition to climate change, increasing human activity (e.g., shipping, mining, fishing, tourism) in the Arctic are new stressors that need to be assessed, as well as other stressors such as global atmospheric pollution and species invasions, which could push the Arctic ecosystems to unknown scenarios (Larsen et al., 2014). All these threats imply rapid changes in the local environments (habitat change) and associated environment connectivity (dispersal mechanisms); both factors will modify the FJL freshwater ecosystems and communities.

Rapid environmental change can lead to a reduction in species richness and diversity (extinction event), which is often followed by a transition to novel communities dominated by generalist species with broader niches (Blois et al., 2013). This scenario represents a possible negative impact on specialized circumpolar taxa, with lower densities in FJL and other Arctic regions. There is a decreasing diversity gradient from the equator to the poles, although it is weaker and patchier for freshwater compared to terrestrial ecosystems (Hillebrand, 2004). Paleolimnological records across the Arctic region indicate that recent climate warming has driven diatom compositional changes with an increase in local diatom richness. These changes have been related to an increase in the length of the growing season (ice-free period), which increase habitat availability and the structural complexity of diatom communities (e.g., development of complex of periphytic communities) (Smol et al., 2005; Lim, Smol & Douglas, 2008). An increase in air temperature, and consequently an increase in the length of the ice-free season (growing period) would facilitate the establishment of species from warmer regions (i.e., Lapland, low tundra in the Canadian Arctic), as has been observed in other ecosystems (Dornelas et al., 2014). The introduction of species would be accelerated by the expected increase in connectivity among Arctic regions due to an increase in the human footprint. In addition, an increase in temporal climate variability (seasonal and inter-annual) could promote species coexistence by reducing competitive exclusion through fluctuation-dependent mechanisms such as storage effect and relative nonlinearities of competition (Chesson, 2000). Our data show that sites with higher diversity are characterized by the presence of slightly acidophilic and aerophilic species (cluster 2), which indicate fluctuating environmental conditions.

However, to sustain local species diversity, the total surface area of available habitat must be maintained. Despite the lower surface area-taxa richness relationship in high latitude regions and in freshwater ecosystems, it is higher for islands (Drakare, Lennon & Hillebrand, 2006). Hence, in the short-term, FJL will increase aquatic freshwater habitats due to melting ice as reported from other similar arctic regions (e.g., E Pinseel et al., unpublished data). In the long-term, the expected decrease of water bodies in the Arctic related to climate warming such as changes in the permafrost (i.e., loss of thermokarst lakes) (Larsen et al., 2014) and a reduction of ice would lower habitat availability, despite the observed pole-ward transport of moisture (Zhang et al., 2012). Consequently, we would expect a short term increase in diversity due to a warmer climate, followed by a decrease due to habitat loss.

At the global scale there are predicted increases in rates of extinction, particularly in the polar regions (Millennium Ecosystem Assessment, 2005). However, there is evidence that biodiversity will remain constant or even increase over time (Vellend et al., 2013; Dornelas et al., 2014), including human impacted areas and isolated oceanic islands (Sax, Gaines & Brown, 2002). This apparent contradiction (species loss versus gain) is due to different scales of study and measures of diversity, which need to be taken into account when considering biodiversity patterns (McGill et al., 2015). For instance, an increasing trend in local α-diversity could parallel a decreasing trend in global β-diversity (Sax & Gaines, 2003; Dornelas et al., 2014; Magurran et al., 2015).

One of the main factors determining β-diversity is regional connectivity and habitat heterogeneity. The general expected increase in human footprint related activities across circumpolar regions (e.g., tourism, polar transport, fishing, and natural resource extraction) would increase the homogenization of communities across the Arctic, therefore reducing β-diversity. Conversely, if drifting sea-ice has been enhancing FJL diatoms α-diversity by facilitating propagule dispersion, then a reduction in sea-ice would increase β-diversity. This change in diversity could be offset by an increase in human visitation. Ice-floe trajectories over FJL archipelago show two main sources: one from the Ob and Yenisei rivers through the Kara Sea, and the other from the Lena River and New Siberian Islands (Novosibirskiye Ostrova Archipelago) through the Láptev Sea (Pfirman et al., 1997). FJL diatoms suggest connections with these northern Siberia areas. The continuous arrival of freshwater diatoms from northern Siberia could also stabilize populations of rare diatom species (low abundance) species, which would otherwise become extinct in a harsh and fluctuating environment.

In the short term, the α-diversity of FJL diatoms is expected to increase with climate warming in association with an increase in human activities. In contrast, β-diversity (circumpolar) is expected to decrease due to an increase in circumpolar connectivity resulting from stressors related to an increase in anthropogenic activities. However, in the longer-term, an expected loss of freshwater bodies and habitat diversity could reduce diatom populations and increase competitive exclusions and therefore local species extinctions (decreasing α-diversity; Fig. 6). This process would be worse for rare species, which are mainly of circumpolar distribution or regionally distributed. We could argue that Arctic species have adapted to past climate fluctuations and will be able to persist. However, the human footprint will compound the impact of isolation due to less sea-ice (extension and duration), which has occurred in past warmer periods, consequently producing a net loss in β-diversity (Fig. 6). Therefore, local α-diversity could not be replenished as fast as in previous periods with the return of a new colder climate period in association with a parallel increase in connectivity across the Arctic Ocean.

Figure 6 Hypothetical models on how species richness changes on two different warming scenarios.

(A) Past warming periods; (a.1) Increasing temperature would promote a rapid increase in FJL α-diversity, but a decrease of circumpolar β-diversity due to biotic homogenization across the arctic. (a.2) Shortly, the lower extension and duration of circumpolar sea-ice would increase isolation among circumpolar regions, and different species would be locally selected. This isolation would reverse the previous decreasing trend in β-diversity. (a.3) However, later on, an expected generalized loss of freshwater ecosystems and the consequent increase in local competitive exclusion would reduce FJL α-diversity. On the end of a warm period cycle (a.4) a net increase on circumpolar diversity (γ-diversity) would be expected by local speciation. (B) Anthropocene warming scenario. Would be expected similar changes on α-diversity, however, the higher expected human frequentation would increase connectivity, which would cancel out the positive effects of isolation on β-diversity. This expected homogenization of circumpolar diatom flora due to higher connectivity and the expected habitat lost (climate warming) could reduce the overall γ-diversity with a net loss of circumpolar species.

Our discussion is founded on the novelty of the recent warming (habitat change), which parallels an increase in Arctic human activity (dispersal facilitation). However changes in other environmental and dispersal factors such as bird migration pathways, and atmospheric and ocean circulation modes could also affect the distribution of circumpolar diatoms. Furthermore, the sensitivity and adaptability of diatoms to environmental change are also dependent on endogenous factors (e.g., genetic diversity, phenotypic plasticity, species traits, populations density, fitness, dispersal ability, persistence, etc.), which are likely to modulate a disparity in diatom responses to external threats and consequently their vulnerability and future distribution (Dawson et al., 2011).

Finally, the low frequency of small araphid species flocks such as Staurosirella, Staurosira, and Pseudostaurosira, also observed in Svalbard (E Pinseel et al., unpublished data), has interesting biogeographic implications. A better distributional record of these taxa across the circumpolar region is required including the need to explore more freshwater habitats in the FJL Archipelago, such as hard substrate habitats, ponds, and stream that are largely influenced by sea bird colonies, as well as other factors, to characterize the complete diatom diversity of this isolated and northern archipelago.

To conclude, there is an urgent need to explore the Arctic to understand, predict, and mitigate future environmental change scenarios to preserve this unique ecosystem. In this regard, monitoring programs and/or paleolimnological approaches using diatoms as environmental indicators could be used as early warnings of environmental changes.

Supplemental Information

Appendix S1 Table of Common FJL diatoms

Common diatoms species (abundance >2% at least in one sample) in FJL with the code used in plots, their geographic distribution, the mean, the maximum and the standard deviation of their relative abundance (percentages) in FJL samples.

Click here for additional data file.

Appendix S2 Plate A SEM imatges of prominent diatom taxa from Franz Joseph Land

Figs. (A–B), Diatoma tenuis, external view. Fig. (C), Merdion circulare, internal view. Fig. (D), Hannaea arcus, internal view. Fig. (E), Diadesmis gallica external view. Fig. (F), internal view. Scale bars. Fig. (D) = 20 µm, Fig. (A) = 10 µm, Figs. (C–D) = 5 µm, Figs. (B, F) = 2 µm.

Click here for additional data file.

Appendix S3 Plate B

Fig. (A), Cymbella cleve-eulerae, external view. Fig. (B). Humidophila ingeiiformis, external view. Fig. (C), Hygropetra balfouriana, external view. Fig. (D), Cymbopleura stauroneiformis, external view. Fig. (E), Cavinula cocconeiformis, external view. Fig. (F), Pinnularia biceps, external view. Scale bars. Fig. (F) = 20 µm, Figs. (A, D) = 10 µm, Fig. (E) = 5 µm, Figs. (B–C) = 2 µm.

Click here for additional data file.

Appendix S4 Plate C

Plate 3. Fig. (A), Nitzschia homburgiensis, external view, Fig. (B), Nitzschia frustulum, external view, Fig. (C), Eunotia scandiorussica, external view, Fig. (D), Eunotia septentrionalis, internal view. Fig. (E), Achnanthidium sp. (Achnanthidium minutissimum sensu lato), external view, Fig. (F), Psammothidium marginulatum, external view. Scale bars. Figs. (A, C) = 10 μ m, Figs. (B, D, E–F) = 5 µm.

Click here for additional data file.

Supplemental Information 1 Diatom counts raw data

Click here for additional data file.

We thank the Russian Arctic National Park and the Ministry of Natural Resources and the Environment of the Russian Federation for their support and providing permits for the expedition. Thanks to the Russian Geographical Society, the Russian Academy of Sciences, Victor Boyarsky for their support and the National Geographic “Pristine Seas Expedition FJL 2013”.

Additional Information and Declarations

Competing Interests

Author Contributions

Field Study Permissions

Data Availability

The authors have no conflict of interest with respect to payment of services by third parties, and any influence or potential influence from organizations and institutions apart from our employers, CREAF, National Park Russian Arctic, CEAB-CSIC, The Canadian Museum of Nature and National Geographic Society. Alan Friedlander and Enric Sala are employees of National Geographic Society, Washington, DC.

Sergi Pla-Rabés conceived and designed the experiments, analyzed the data, contributed reagents/materials/analysis tools, wrote the paper, prepared figures and/or tables, reviewed drafts of the paper.

Paul B. Hamilton conceived and designed the experiments, analyzed the data, contributed reagents/materials/analysis tools, prepared figures and/or tables, reviewed drafts of the paper.

Enric Ballesteros conceived and designed the experiments, contributed reagents/materials/analysis tools, reviewed drafts of the paper, sampling, and fieldwork logistics.

Maria Gavrilo conceived and designed the experiments, prepared figures and/or tables, reviewed drafts of the paper, sampling, and fieldwork logistics.

Alan M. Friedlander and Enric Sala conceived and designed the experiments, reviewed drafts of the paper, sampling, and fieldwork logistics.

The following information was supplied relating to field study approvals (i.e., approving body and any reference numbers):

Approval to conduct field studies was granted by the Russian Federation Ministry of Education and Science Approval Ref. No. 14-368 of 06.05.2013 permission to the Russian Arctic National Park.

The following information was supplied regarding data availability:

Franz-Josef Land Diatom Counts.

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
