# Peer review of "The structure and diversity of freshwater diatom assemblages from Franz Josef Land Archipelago: a northern outpost for freshwater diatoms"

_PeerJ, doi:10.7717/peerj.1705_

## Round 0.1 · original submission · Minor Revisions

Dear Authors

Please address the requested minor revisions and submit your revised Mss.

Regards,

Reviewer 1 ·

Basic reporting

See general comments section

Experimental design

No comments

Validity of the findings

See general comments section

Additional comments

The manuscript presents new data on important bioindicators of environmental change (ie, diatoms) from a previously little studied region. It warrants publication, but several issues need to be addressed. See below.

Line 27, 35, 38, 140, 147: Watch the plural of diatom assemblages/diversity/identifications. It should not read as “diatoms”

Line 32: taxon, not taxa

Line 43: Suggest changing to – resilience of this region and its diversity

Line 59: It’s unclear what is meant by “stability of these freshwaters systems”. Does this refer to species changes with warming? Ponds completely desiccating? Please elaborate.

Line 118: Baffin Island is not the Canadian High Arctic. Where is this temperature data from? If Iqaluit = that is below the Arctic Circle. The high Arctic of Canada is usually considered to be islands above Baffin/Lancaster Sound.

Line 125, 173: How many sites were sampled? It is reported that 18 were sampled, but 2 lakes, 6 ponds, and 8 streams = 16 total. Please clarify.
Also in Fig. 4, there are 9 circles denoting ponds, but in methods it says only 6 ponds were sampled?

Line 243: Given that only rock scrapes were sampled, do the authors feel that the total number of taxa were underestimated in this present study compared to previous surveys, which typically use sediment samples and therefore integrate more habitats within each site? This might be worth addressing

Line 253: I think you mean “sites”, not samples.

Line 272: Was any water chemistry recorded? pH values? If not, it seems speculative to assume there is a pH gradient without direct measurements. Is geology very different? Lake age (and till exposure) is given as a possible explanation, but did this region not deglaciate at around the same time?

Line 288: Please elaborate what is meant by “diatom assemblages showing higher resilience”. Higher resilience to what?

Line 290: The explanation of some ponds showing greater variability in terms of desiccation due to the presence of aerophilic taxa would be ok if sediment samples were analyzed, but not necessarily rock scrapes. Sediment samples cover the last several years/decades of accumulation, whereas rock scrapes should only contain diatom assemblages of the present year due to the scouring effects of ice. When the rock scrapes were taken, where in the pond were they recovered from? Underneath water? If underwater, (and assuming the rock scrapes represent only one year of diatom growth) these taxa, even if commonly identified as aerophiles, cannot be indicating desiccation of the site. In summary, to infer any sort of environmental variability at any site, you would need the collection of diatom assemblages from several years or decades, not just a single growing season.

Line 318: Why are species indicative of moss and sediment microhabitats found on your rock scrapes? Perhaps they are more diverse wrt habitat than previously thought?

Line 349-354: So rather than the concept of circumpolar taxa, it is more apt to refer to them as cold environment assemblages?

Line 356: What is a “zakaznik”?

The last section on “Future scenarios of diversity” is rather long and speculative. Oddly there is no mention of paleolimnological studies from other Arctic regions to draw comparisons as to how diatom assemblages will respond to climatic variations of the future. Also, there was no mention of changes to growing season length, or pond desiccation as evapotranspiration increases. These are arguably the 2 most important climate-related factors that can influence diatom assemblages in the Arctic, rather than say tourism and ice rafting etc. Also, with a section this long the authors say little about any implications of a climate-related change in species diversity.

Fig 2. Indicator is misspelled

Fig. 3 Evenness is misspelled

Reviewer 2 ·

Basic reporting

No major comments, minor suggestions are reported below

Experimental design

No major comments, minor suggestions are reported below

Validity of the findings

I do not see any strong evidence how exploration and tourism would be able to decrease the biodiversity. An increase of habitats and deposition of e.g. nutrients would rather lead to an aquatic biodiversity increase. But, is biodiversity really a good management goal for this region? Or should these environments rather be protected to preserve the “low” biodiversity? The mechanisms of biodiversity change on FJL are to some extent speculative, and not fully really relevant. I suggest that the author reconsider the reasoning of this part of the manuscript.

Additional comments

The manuscript investigates the diatom distribution in the Franz Josef Land archipelago (FJL), analyzing epilithic material from streams and sediments from ponds. Biodiversity patterns are analyzed in the light of different habitats and diatom classifications. A conceptual model is presented how environmental changes (climate warming) and increasing exploitation could affect future biodiversity.
The study as such is rather descriptive, but succeeds to evaluate the findings in the light of historical data form the area, and also recent studies from circumpolar Arctic regions.
In general, the manuscript is of high quality. However, the role of some drivers of biodiversity remain unclear, and should be reconsidered (detailed comments below).

Minor comments
• Reconsider the third sentence in the abstract (“Cosmopolitan…). The classification of diatom taxa is too arbitrary to support such a general statement (lines 29-31).
• What evidence do you have that lakes deeper than 1 meter do not freeze to the bottom? This boundary seems arbitrary (lines 120-121).
• Explained variance numbers in the text are not identical as in Figure 4 (lines 201-218).
• Is color really needed for Figure 1? I don’t think coloring the figure adds any necessary information.

Language
• diatom assemblages, and not diatoms assemblages (line 27, 35)
• hyphen and dash are not applied consistently throughout the entire manuscript
• habitat loss instead of habitat lost (line 395)
• caption Figure 2, “with higher abundance”, please be more specific (first line)
• caption Figure 2, sorted instead of sorter (second line)
• Figure 3, Evenness instead of Evensess (title panel 4 and 5)
• Figure 4, percentage values for PCO axis 1 and 2 are not according to the text
• Figure 4, ordinate label should read PCO axis 2 (and not 1)
• Figure 5, Evenness instead of Evensess (title panel 4 and 5)

References
• AMAP, year missing (line 49)
• Van der Vijver et al. 2004 (line 142)
• Reference to Figure 5 instead of figure 6 (line 235)
• Foged 1980? Probably just a wrong year, alternatively not cited in the reference list (line 248)
• Pinseel et al., submitted or in press? (line 326, 624 and other places)
• Nansen, remove first name (Fridtjof), and only 1897 is in the reference list (not 1906) (line 335)
• Cleve 1899 should read 1898 (line 338)
• Typing error (dots) (line 554)
• Issue and pages missing (line 562)
• Lange-Bertalot (line 577)

---

## Round 0.2 · accepted · Accept

The revised manuscript is up to the standard of the journal. I recommend the revised article for publication in PeerJ.